# Electromotive force in the solar wind

Yasuhito Narita[1]

[1]Space Research Institute, Austrian Academy of Sciences, Schmiedlstr. 6, A-8042 Graz, Austria

**Correspondence:** Y. Narita
(yasuhito.narita@oeaw.ac.at)

**Abstract.** The concept of electromotive force appears in various electromagnetic applications in geophysical and astrophysical fluids. A review of the electromotive force and its applications to the solar wind are discussed such as the electromotive force profile during the shock crossings and the observational tests for the mean-field model against the solar wind data. The electromotive force is being recognized as serving as a useful tool to construct a more complete picture of space plasma turbulence when combined with the energy spectra and helicity profiles.

## 1 Introduction

Electromotive force is one of the electric field realizations in electrically conducting fluids or plasmas, and is excited by turbulent fluctuations of flow velocity and magnetic field on smaller spatial or temporal scales. The electromotive force plays an essential role in the dynamo mechanism in which the large-scale magnetic field is generated by amplifying small-scale magnetic fields in turbulent fluid motions (Elsasser, 1956; Moffatt, 1978; Roberts and Soward, 1992). Examples of large-scale magnetic field generation associated with the dynamo mechanism can be found in geophysical, solar system, and astrophysical applications such as Earth's magnetic field (Glatzmaier and Roberts, 1998; Glatzmaier, 2002; Roberts and Glatzmaier, 2000; Kono and Roberts, 2002), planetary magnetic fields (Jones, 2011), Jupiter's moon (Ganymede) intrinsic field (Schubert et al., 1996; Sarson et al., 1997), solar magnetic field (Charbonneau, 2010, 2014; Brandenburg, 2018), stellar magnetic fields (Berdyugina, 2005; Brun and Browning, 2017), and galactic and extragalactic fields (Vainshtein and Ruzmaikin, 1971; Kronberg, 1994; Widrow, 2002; Beck et al., 2020). Our understanding of the dynamo mechanism is being deepened and broadened by using numerical simulations using the fundamental equations and analytic treatment and modeling (Brandenburg, 2018). Recent theoretical study by Yokoi (2018a) suggests that the electromotive force and the density variation are locally enhanced such as in the shock-front region, and the density enhancement would lead to a fast magnetic reconnection.

Along with the advent of the inner heliospheric missions such as Parker Solar Probe (Fox et al., 2016) and Solar Orbiter (Müller et al., 2020), the concept of electromotive force is being re-introduced in the field of space plasma physics after pioneering works by Marsch and Tu (1992, 1993). In particular, it is found that the electromotive force computed from the Helios spacecraft data in the solar wind becomes locally enhanced during the magnetic cloud or shock crossing in interplanetary space (Bourdin et al., 2018; Narita and Vörös, 2018; Hofer and Bourdin, 2019).

This article presents a review of the electromotive force studies in the solar wind in view of the current in situ observations in the inner heliosphere such as Parker Solar Probe (since 2018), Solar Orbiter (since 2020), and BepiColombo's cruising

to the Mercury orbit (since 2018) (Benkhoff et al., 2010; Mangano et al., 2021). The theoretical treatment of electromotive force is first introduced (section 2) and the applications (though the number of literatures is limited) to the solar wind are presented (section 3). The article concludes with summary and outlook (section 4). The concept of electromotive force can be

implemented in the spacecraft data in order to construct a more complete picture of the turbulent fluctuations in the solar wind, and has the potential to fill the gap between the processes in the dynamo mechanism in the conducting fluids and turbulence in collisionless space plasmas.

## 2 Theoretical background

The electromotive force is defined as the averaged vector product between the fluctuating flow velocity $\boldsymbol{u}$ and the fluctuating
magnetic field $\boldsymbol{b}$,

$$\boldsymbol{E}_{\mathrm{emf}} \quad = \quad \langle \boldsymbol{u} \times \boldsymbol{b} \rangle. \tag{1}$$

Note that the electromotive force is expressed in units of the electric field $[\mathrm{V/m}]$. Derivation of Eq. 1 is as follows. We apply the decomposition of the magnetic field and flow velocity into the mean or large-scale fields ($\langle \boldsymbol{B} \rangle$ and $\langle \boldsymbol{U} \rangle$) and the fluctuating fields as

$$\boldsymbol{B} \quad = \quad \langle \boldsymbol{B} \rangle + \boldsymbol{b} \tag{2}$$
$$\boldsymbol{U} \quad = \quad \langle \boldsymbol{U} \rangle + \boldsymbol{u}, \tag{3}$$

where the angular bracket $\langle \cdots \rangle$ denotes the operation of statistical averaging or smoothing. The fluctuating fields $\boldsymbol{b}$ and $\boldsymbol{u}$ have vanishing mean values, $\langle \boldsymbol{b} \rangle = 0$ and $\langle \boldsymbol{u} \rangle = 0$, but the average of a product of fluctuating fields does not vanish, for example, the energy density of the fluctuating magnetic field is $(2\mu_0)^{-1}\langle \boldsymbol{b} \cdot \boldsymbol{b} \rangle$, where $\mu_0$ is the permeability of free space. The electromotive
force arrises when the mean field picture is applied to the induction equation in magnetohydrodynamics,

$$\partial_t \langle \boldsymbol{B} \rangle \quad = \quad \nabla \times (\langle \boldsymbol{U} \rangle \times \langle \boldsymbol{B} \rangle) + \nabla \times \langle \boldsymbol{u} \times \boldsymbol{b} \rangle + \eta \nabla^2 \langle \boldsymbol{B} \rangle. \tag{4}$$

Here, $\eta$ is the magnetic diffusivity, which is related to the conductivity $\sigma$ through $(\mu_0\sigma)^{-1}$. The first term on the right hand side in Eq. (4) represents frozen-in of the large-scale magnetic field (strictly speaking, deformation of the large-scale magnetic field by the large-scale flow), the second term represents the curl of electromotive force, and the third term represents the
50 diffusion of large-scale field. The electromotive force can act both as constructive to the large-scale field (e.g., amplification of large-scale field by fluctuations such as in the dynamo mechanism) and as destructive (e.g., scattering or disturbance of large-scale field by fluctuations such as in plasma turbulence) The electromotive force is one of the second-order fluctuation quantities and is closely related to the concept of energy densities (magnetic energy; kinetic energy) and helicity densities (cross helicity; current helicity; kinetic helicity). Magnetic helicity, for example, describes the three-dimensional topological
properties of magnetic field lines (Berger and Field, 1984; Berger, 1999). Helical structures also play an important role in fluid dynamics (Moffatt, 2014). The build-up of large-scale magnetic field in a helical flow is demonstrated using a semi-analytic treatment of magnetohydrodynamic turbulence (Pouquet et al., 1976).

The electromotive force can be observationally determined when the flow velocity data and the magnetic field data are available. In general, in the observational studies, it is more practical to construct the covariance matrices for the magnetic field as $\mathbf{M_{bb}}$, for the flow velocity as $\mathbf{M_{uu}}$, and for the cross correlation between the flow velocity and the magnetic field as $\mathbf{M_{ub}}$. The electromotive force is constructed from the off-diagonal elements of the cross correlation matrix $\mathbf{M_{ub}}$. The mean-field dynamo theory predicts that the electromotive force is related to the energy and helicity quantities. Magnetic energy corresponds to the diagonal elements of the matrix $\mathbf{M_{bb}}$, and the kinetic energy corresponds to the diagonal elements of the matrix $\mathbf{M_{uu}}$. Magnetic helicity and current helicity are constructed from the off-diagonal elements of the magnetic field matrix $\mathbf{M_{bb}}$, and the kinetic helicity from the off-diagonal elements of the flow velocity matrix $\mathbf{M_{uu}}$. The cross helicity is constructed from the diagonal elements of the cross correlation matrix $\mathbf{M_{ub}}$. The appendix section shows the second-order quantities that are accessible to the spacecraft observations.

Amplification and scattering of the large-scale field by fluctuating fields are formulated in the turbulent dynamo mechanism by associating the electromotive force with the large-scale field and its spatial derivatives to close the equations for the large-scale fields. A simpler yet symmetric (with respect to the curl of magnetic field and that of flow velocity) form is proposed from the study of reversed field pinch (Yoshizawa, 1990) and cross helicity dynamo (Yokoi, 2013) as

$$\langle \boldsymbol{u} \times \boldsymbol{b} \rangle = \alpha \langle \boldsymbol{B} \rangle - \beta \nabla \times \langle \boldsymbol{B} \rangle + \gamma \nabla \times \langle \boldsymbol{U} \rangle. \tag{5}$$

The first term with the coefficient $\alpha$ represents amplification of the large-scale magnetic field by helical flow motions (cf. alpha dynamo mechanism). The second term with the coefficient $\beta$ represents scattering of the large-scale field by turbulent fluctuations. In fact, the $\beta$ term yields $\beta \nabla^2 \langle \boldsymbol{B} \rangle$ in the induction equation, which is identified as turbulent diffusion of the large-scale field. The third term with the coefficient $\gamma$ represents amplification of the large-scale field (and hence leading to a type of dynamo mechanism) by the non-zero cross helicity effect. It is important to note that the association of electromotive force with the large-scale fields is an assumption, and its validity needs to be studied by, e.g., numerical simulations, laboratory experiments, or in situ measurements in space. With the Ansatz in Eq. 5, the induction equation for the large-scale field (Eq. 4) has amplification and turbulent diffusion terms explicitly:

$$\partial_t \langle \boldsymbol{B} \rangle = \nabla \times (\langle \boldsymbol{U} \rangle \times \langle \boldsymbol{B} \rangle) + \nabla \times (\alpha \langle \boldsymbol{B} \rangle + \gamma \nabla \times \langle \boldsymbol{U} \rangle)$$
$$+ (\beta + \eta) \nabla^2 \langle \boldsymbol{B} \rangle. \tag{6}$$

The coefficient $\alpha$ represents the strength of the kinetic helicity (a measure of helical flow) and the coefficient $\beta$ represents the turbulent diffusion. Practical forms of the transport coefficients $\alpha$ and $\beta$ are, after Steenbeck and Rädler (1966) and Krause and Rädler (1980), expressed as

$$\alpha = -\frac{1}{3} \tau \langle \boldsymbol{u} \cdot (\nabla \times \boldsymbol{u}) \rangle \tag{7}$$

$$\beta = \frac{1}{3} \tau \langle \boldsymbol{u} \cdot \boldsymbol{u} \rangle, \tag{8}$$

with a proper time scale $\tau$ (turbulence correlation time). which needs to be evaluated separately using some turbulence model (e.g., eddy turnover time). The coefficient $\gamma$ is modeled, in analogy to the coefficients $\alpha$ and $\beta$, after Bourdin et al. (2018), as

$$\gamma \quad = \quad \frac{1}{3}\tau \langle \boldsymbol{u} \cdot \boldsymbol{b} \rangle \tag{9}$$

More comprehensive forms of the transport coefficients are presented in view of cross helicity dynamo (Hamba, 1992; Yoshizawa, 1998; Yokoi and Balarac, 2011; Yokoi, 2013, 2018b) as

$$\alpha \quad = \quad C_\alpha \tau \left\langle -\boldsymbol{u} \cdot (\nabla \times \boldsymbol{u}) - \frac{\boldsymbol{b}}{\sqrt{\mu_0 \rho}} \cdot \left( \nabla \times \frac{\boldsymbol{b}}{\sqrt{\mu_0 \rho}} \right) \right\rangle \tag{10}$$

$$\beta \quad = \quad C_\beta \frac{\tau}{2} \left\langle |\boldsymbol{u}|^2 + \frac{1}{\mu_0 \rho}|\boldsymbol{b}|^2 \right\rangle \tag{11}$$

$$\gamma \quad = \quad C_\gamma \tau \langle \boldsymbol{u} \cdot \boldsymbol{b} \rangle . \tag{12}$$

with $C_\alpha = O(10^{-2})$, $C_\beta = O(10^{-1})$ and $C_\gamma = O(10^{-1})$.

It is worth noting that the assumptions in the derivation of the transport coefficients are different between Eqs. (7)–(8) and Eqs. (10)–(12); the former expressions are based on homogeneous turbulence in a rotating flow, while the lattere expressions are based on the response function (Green function) of inhomogeneous turbulence. Extension of Eq. (7) to Eq. (10) indicates that the residual helicity between the kinetic helicity and the current helicity drives the dynamo effect (Pouquet et al., 1976). The importance of the cross helicity term (with the coefficient gamma) has largely been overlooked in the earlier studies because the large-scale flow velocity was eliminated by using the Galilean invariance.

Transport of the kinetic helicity and the current helicity (or magnetic helicity) from the solar convection zone to the heliosphere remains one of the open questions. The spacecraft observations indicate that magnetic helicity changes the sign nearly randomly over the spacecraft-frequencies (Matthaeus et al., 1982). However, as discussed in section 3, the alpha effect may locally be enhanced when a transient event (e.g., coronal mass ejections) passes by.

Diffusion of large-scale magnetic field by the beta term is expected to be persistently large in the solar wind, considering the fact that the solar wind exhibits sign of developed or fully-developed turbulence with power-law energy spectra for the flow velocity and the magnetic field.

The cross helicity effect may play an important role in the solar wind, as the cross helicity can be interpreted as the energy difference between two counter-propagating Alfvén wave packets when using the Elsässer variables, and is expected to evolve in the solar wind over the heliocentric distances if the Alfvén waves are excited near the Sun, propagate uni-directionally (away from the Sun) in the inner heliosphere, and gradually undergoes scattering or instabilities to excite backward-propagating Alfvén waves.

## 3 Applications in the solar wind

### 3.1 Overview

In the observational studies, the electromotive force is computed as the cross product of the fluctuating flow velocity and fluctuating magnetic field, and represents the second-order fluctuation quantity. The units of electromotive force can be represented

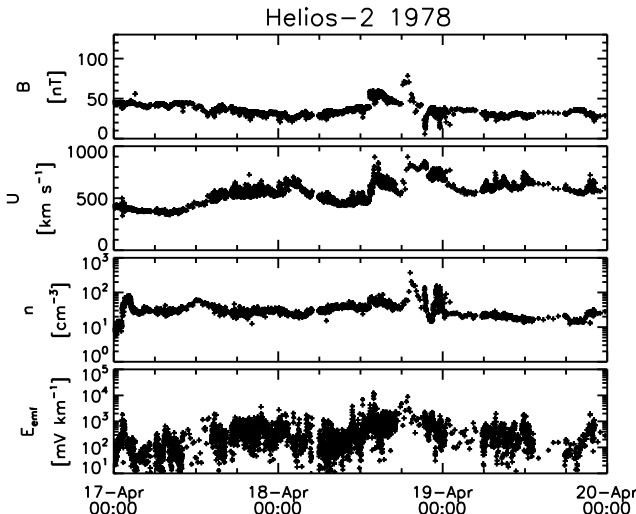

**Figure 1.** Time series plots of magnetic field magnitude, flow velocity, proton number density, and instantaneous electromotive force (without statistical averaging or smoothing) obtained by Helios-2 spacecraft. Note a magnetic cloud or shock crossing at about 1800-1900 UT on 19 April, 1978. Figure is produced using the data after Narita and Vörös (2018).

in units of electric field as follows,

$$[\boldsymbol{u} \times \boldsymbol{b}] \quad = \quad \mathrm{km\,s^{-1}\,nT} \tag{13}$$

$$= \quad \mathrm{mV\,km^{-1}} \tag{14}$$

when using the induction equation relating the electric field to the magnetic field that the ratio of electric to magnetic field has a dimension of velocity.

One of the applications of the electromotive force is diagnosis of plasma and magnetic field dynamics across transient events in the solar wind (e.g., magnetic clouds, coronal mass ejections, co-rotating interaction regions). Both magnetic field amplification (through the alpha term) and turbulent diffusion (the beta term) are locally enhanced during the transient events, suggesting that the solar wind serves as a natural laboratory for testing for the dynamo theory.

An example of electromotive force profile is displayed in Fig. 1. The magnetic field and ion measurements by the Helios-2 spacecraft are used to compute the electromotive force for a quiet solar wind interval (on 17 April, 1978), a magnetic cloud (or shock crossing) event on 18 April, 1978, and a post-shock interval on 19 April, 1978. Electromotive force has different levels of activity, and varies between $10\,\mathrm{mV\,km^{-1}}$ (quiet solar wind) and $10\,\mathrm{V\,km^{-1}}$ (magnetic cloud or shock crossing).

### 3.2 Spectral feature

The electromotive force has nearly random fluctuations as shown in Fig. 1, but the fluctuations are not Gaussian but rather exhibit a turbulence-like power-law energy spectrum. Figure 2 exhibits a spectrum of the out-of-ecliptic component of electro-

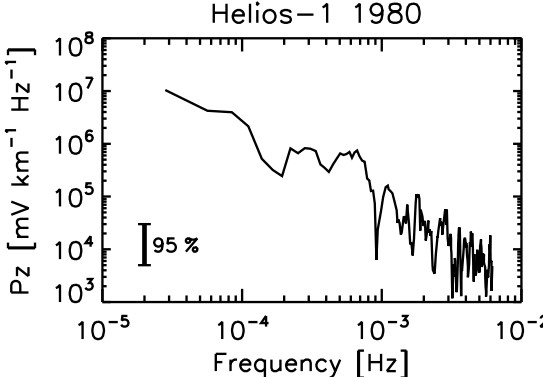

**Figure 2.** Frequency spectrum (in the spacecraft frame) of the out-of-ecliptic component (the z component) of electromotive force derived from the Helios-1 at a radial distance of 0.53 AU, after the spectral data presented by Marsch and Tu (1992). The magnitude value of the electromotive force is plotted here.

motive force $E_{\mathrm{emf,z}}$ as a function of the spacecraft-frame frequencies after Marsch and Tu (1992, 1993). The magnetic field and ion data obtained by the Helios-1 spacecraft at a distance of about 0.53 AU in 1980 are used to compute the electromotive force spectrum.

The solar wind speed is about 637 km s$^{-1}$ on the analyzed time interval, and the fluctuations are highly Alfvénic (with the components propagating away from the Sun dominating the fluctuations) and the energy spectrum is close to Kolmogorov's inertial-range spectrum with a slope of $-5/3$. The frequency spectrum may thus be regarded as nearly streamwise wavenumber spectrum when Taylor's frozen-in flow hypothesis is used. The electromotive force vanishes in the purely Alfvénic fluctuations, since the fluctuating flow velocity is either positively or negatively correlated to the fluctuating magnetic field. The overall power-law spectral formation is indicative of some turbulent cascade mechanism operating in the electromotive force.

### 3.3 Observational tests

#### 3.3.1 Test for the alpha effect

The validity of mean-field model can be tested against solar wind data in various ways. Marsch and Tu (1992) regarded the mean field model as a Taylor expansion with respect to the mean magnetic field $\langle \boldsymbol{B} \rangle$ as the leading order and and its spatial gradients (or curl of mean magnetic field) $\nabla \times \langle \boldsymbol{B} \rangle$ as higher-order terms. If the large-scale current is in the direction to the mean magnetic field (force-free configuration for the large-scale fields), and if the cross helicity term (with the coefficient $\gamma$) is negligible, the electromotive force is proportional to the mean magnetic field

$$\boldsymbol{E}_{\mathrm{emf}} \quad \propto \quad \alpha \langle \boldsymbol{B} \rangle. \tag{15}$$

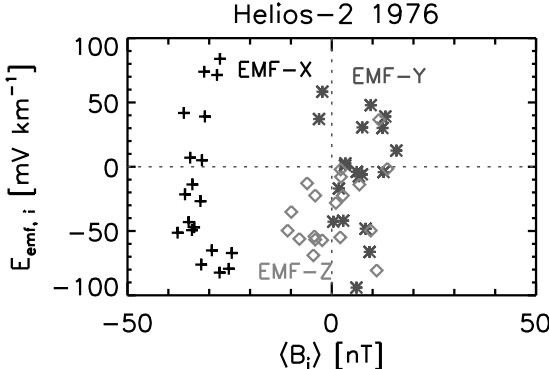

**Figure 3.** Test for the alpha effect by plotting the electromotive force $\boldsymbol{E}_{\mathrm{emf}}$ as a function of the mean magnetic field $\langle \boldsymbol{B} \rangle$ using the Helios-2 solar wind data near perihelion (0.29 AU from the Sun) in 1976. The x component (EMF-X) is radially away from the Sun, the y component (EMF-Y) is azimuthally westward (with respect to the ecliptic north), and the z component (EMF-Z) is northward to the ecliptic plane. Figure is produced using the data set presented by Marsch and Tu (1992).

The simplified model (Eq. 15) is tested against the Helios-2 observation of fast solar wind near 0.29 AU in 1976. Figure 3 displays a scatter plot of the electromotive force as a function of the mean magnetic field for three components: radially outward direction from the Sun (the x component) with plus signs in black, azimuthally westward direction (the y component) with asterisk symbols in dark gray, and solar-ecliptic north direction (the z component) with diamond symbols in light gray.

The alpha effect test result (Fig. 3) shows that no clear correlation is observed between the electromotive force and the mean magnetic field. The scatter is larger in the electromotive force than that in the mean field.

### 3.3.2 Evaluation of the alpha and beta coefficients

For the simple model with the alpha and beta terms (indicating the field amplification and the turbulent diffusion, respectively), analytic forms are proposed to estimate the transport coefficients alpha and beta (Narita and Vörös, 2018). For this purpose we model the electromotive force in the following form:

$$\boldsymbol{E}_{\mathrm{emf}} = \alpha \langle \boldsymbol{B} \rangle - \beta \nabla \times \langle \boldsymbol{B} \rangle. \tag{16}$$

Vector product between the mean magnetic field $\langle \boldsymbol{B} \rangle$ and the electromotive force $\boldsymbol{E}_{\mathrm{emf}}$ in Eq. (16) yields

$$\langle \boldsymbol{B} \rangle \times \boldsymbol{E}_{\mathrm{emf}} = -\beta \langle \boldsymbol{B} \rangle \times (\nabla \times \langle \boldsymbol{B} \rangle), \tag{17}$$

which can be arranged into an estimator for the beta coefficient as

$$\beta = \frac{1}{F^2} \boldsymbol{F} \cdot (\langle \boldsymbol{B} \rangle \times \boldsymbol{E}_{\mathrm{emf}}). \tag{18}$$

Here, $\boldsymbol{F}$ denotes the Lorentz force for the large-scale magnetic field and is defined as (by setting the permeability of free space $\mu_0$ to unity for simplicity)

$$\boldsymbol{F} = (\nabla \times \langle \boldsymbol{B} \rangle) \times \langle \boldsymbol{B} \rangle. \tag{19}$$

For the coefficient alpha we multiply Eq. (16) by the mean magnetic field $\langle \boldsymbol{B} \rangle$ and obtain:

$$\langle \boldsymbol{B} \rangle \cdot \boldsymbol{E}_{\mathrm{emf}} = \alpha(\langle \boldsymbol{B} \rangle)^2 - \beta \langle \boldsymbol{B} \rangle \cdot (\nabla \times \langle \boldsymbol{B} \rangle). \tag{20}$$

Equation (20) can be arranged into:

$$\alpha = \frac{1}{\langle \boldsymbol{B} \rangle^2} \left[ \langle \boldsymbol{B} \rangle \cdot \boldsymbol{E}_{\mathrm{emf}} + \frac{h_{\mathrm{crt}}}{F^2} \boldsymbol{F} \cdot (\langle \boldsymbol{B} \rangle \times \boldsymbol{E}_{\mathrm{emf}}) \right], \tag{21}$$

by using the estimator for the coefficient beta (Eq. 18) and introducing the large-scale current helicity density $h_{\mathrm{crt}}$ as

$$h_{\mathrm{crt}} = (\nabla \times \langle \boldsymbol{B} \rangle) \cdot \langle \boldsymbol{B} \rangle. \tag{22}$$

The coefficients alpha and beta are evaluated observationally using Eqs. (18) and (21), and graphically plotted as functions of the fluctuating flow speed $u = |\boldsymbol{u}|$ and fluctuating magnetic field $b = |\boldsymbol{b}|$ on the logarithmic scale in Fig. 4. The coefficients alpha and beta exhibit the following properties:

1. Both the coefficients are scattered to a larger extent over the flow speed fluctuation $u$ and the magnetic field fluctuation $b$. Variation of the coefficient alpha spans from $10^{-4}$ km s$^{-1}$ to $10^4$ km s$^{-1}$ (8 orders of magnitude), and that of beta span from $10^6$ km$^2$ s$^{-1}$ to $10^{16}$ km$^2$ s$^{-1}$ (10 orders of magnitude).

2. Yet, both the coefficients show a systematic trend that the values of coefficients increase at larger fluctuation amplitudes. The systematic trend appears not only in the flow speed domain (left panels) but also in the magnetic field domain 185 (right panels). The systematic trend may as well be (observationally) modeled using a power-law scaling (linearly on the logarithmic scale).

### 3.3.3  Test for the mean-field model

The electromotive force can be evaluated by directly computing the cross product between the fluctuating flow velocity and the fluctuating magnetic field after Eq. (1) and also by making use of the mean-field model with the helical dynamo term (the 190 alpha term), the magnetic diffusion term (the beta term), and the cross helicity dynamo term (the gamma term) after Eq. (5). By doing so, it is possible to validate the mean-field model using in situ plasma and magnetic field measurements in the solar wind.

Figure 5 displays the time series plot of electromotive force using the Helios-2 observation of magnetic cloud (or shock crossing) event on 18 April, 1976 (the same event as shown in Fig. 1). The electromotive force is then computed with Eq. (5) 195 by estimating the kinetic helicity, magnetic fluctuation energy, and cross helicity, and turbulence correlation time (shown by the

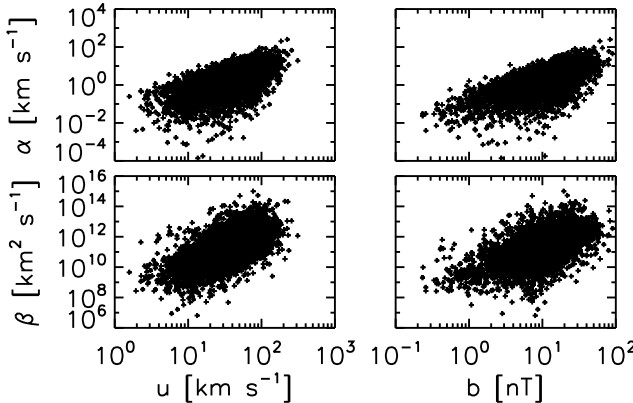

**Figure 4.** Transport coefficients alpha and beta as functions of the fluctuating flow speed and fluctuating magnetic field for the two-component electromotive force model with the alpha and beta terms. The Helios solar wind data and the transport coefficients studied by Narita and Vörös (2018) are used for the graphics.

curve in gray). Though not exact, the mean-field model can qualitatively the observationally-determined electromotive force in the sense that both the peak time and the peak value are in good agreement.

It is interesting to note that the test for the single alpha effect (i.e., proportionality of electromotive force solely to the mean magnetic field without the beta and the gamma effects) fails against the solar wind data after by Marsch and Tu (1992), yet that the test for the model with the three terms including the alpha, beta, and gamma terms successfully reproduces the measured electromotive force after Bourdin et al. (2018). The scaling analysis using Eq. (23) indicates that the alpha term should be almost as important as the beta term (in fact, 4 times larger) which is as important as the gamma term. Hence, the lesson is that the simplest model using only the alpha term is not sufficient, and that the magnetic diffusion and the cross helicity effect should be considered as well in the electromotive force composition. Under which conditions the alpha effect will dominate remains an observationally open question; perhaps there is a dependence on, e.g., fast or slow solar wind, quieter or more disturbed solar wind, association with transient events such as coronal mass ejections and corotating interaction regions.

It is interesting to compare among the three terms in the electromotive force model (alpha-term, beta-term, and gamma-term) in Eq. (5) using the order-of-magnitude estimate method. The reconstruction work by Bourdin et al. (2018) determined the values of coefficients alpha, beta, and gamma as shown in Tab. 1. The ratio of the alpha term (helical dynamo term) to the beta term (turbulent diffusion term) is estimated nearly of the order of unity:

$$\frac{|\alpha\langle\boldsymbol{B}\rangle|}{|\beta\nabla\times\langle\boldsymbol{B}\rangle|} \sim \frac{\alpha L}{\beta} \sim 4 \tag{23}$$

where the spatial gradient scale is estimated about $L = 4 \times 10^6$ km in the solar wind corresponding to a Doppler-shifted frequency of about $10^4$ Hz (e.g., Tu and Marsch, 1995). The order-of-unity estimate as in Eq. (23) is valid for both the active solar wind and the quiet solar wind when referring to the observational values of the transport coefficients in Tab. 1. The ratio

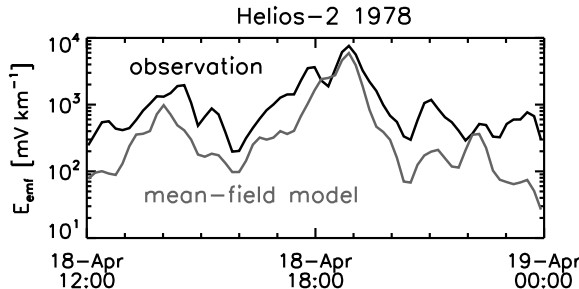

**Figure 5.** Comparison of the electromotive force magnitude computed from the Helios-2 of plasma and magnetic field fluctuation data (in black) and that from the mean-field model using the Helios-2 mean field data (in gray) for the shock crossing event on 18 April, 1978 (the same event as that in Fig. 1). Figure is produced using the data in Bourdin et al. (2018).

**Table 1.** Transport coefficients estimated from a 12-hour solar wind interval including an interplanetary shock (active solar wind) and a quasi-stationary turbulent state (quiet solar wind) after Bourdin et al. (2018).

| coefficient | active solar wind | quiet solar wind |
|:---:|:---:|:---:|
| $\alpha$ | $-50 \text{ km s}^{-1}$ | $\pm 5 \text{ km s}^{-1}$ |
| $\beta$ | $50 \times 10^6 \text{ km}^2 \text{ s}^{-1}$ | $5 \times 10^6 \text{ km}^2 \text{ s}^{-1}$ |
| $\gamma$ | $-10 \times 10^6 \text{ km nT}$ | $\pm 1 \times 10^6 \text{ km nT}$ |

of the gamma term (cross helicity term) to the beta term is estimated of the order of unity, too:

$$\frac{|\gamma \nabla \times \langle \boldsymbol{U} \rangle|}{|\beta \nabla \times \langle \boldsymbol{B} \rangle|} \sim \frac{\gamma B_0}{\beta U_0} \sim 2. \tag{24}$$

Here we used a flow speed of $U_0 = 400 \text{ km s}^{-1}$ (typical both in the inner heliosphere and around the Earth orbit) and a magnetic field of $B_0 = 40$ nT (typical in the inner heliosphere but not around the Earth). The cross helicity term plays a more important role because the flow speed does not change very much over the radial distances from the Sun while the magnetic field decays radially due to the flux conservation over the spatial expansion. Around the Earth orbit, the ratio of the gamma term to the beta term is expected about 10 times larger than that in the inner heliosphere.

### 3.4 Radial evolution in the heliosphere

The electromotive force becomes enhanced during shock crossings, reaching the order of 1 V km$^{-1}$. The spatial distribution or the radial profile of electromotive force during the shock crossings is determined using the whole Helios data in the inner heliosphere down to the perihelion of about 0.29 AU. A shock detection algorithm was developed using the electromotive force, and the algorithm was applied to the whole Helios data to identify 531 shock crossing events (Hofer and Bourdin, 2019).

Figure 6 displays a scatter plot of electromotive force during the shock crossings as a function of the radial distance of observation from the Sun. The shock-enhanced electromotive force a tendency of decay at larger distances from the Sun. The

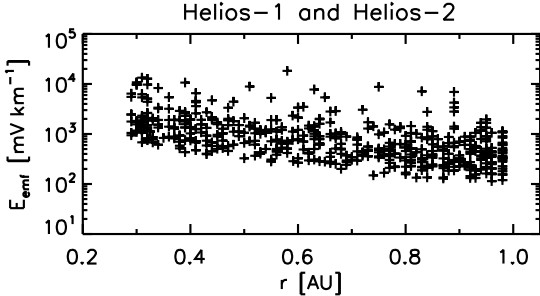

**Figure 6.** Distribution of peak values of electromotive force (as a magnitude) as a function of radial distance from the Sun for shock crossing events in the Helios-1 and Helios-2 data. Figure is produced using the data in Hofer and Bourdin (2019).

electromotive force is 1–10 V km$^{-1}$ near the perihelion (0.29 AU), and decays to 0.1–10 V km$^{-1}$ near the apohelion (close to 1 AU). The spatial decay or radial decay of electromotive force can in practice be fitted by a power-law curve as $r^{-1.54}$ (Hofer and Bourdin, 2019).

## 4   Summary and outlook

The electromotive force has largely been overlooked in space plasma studies in contrast to the conventional turbulence analysis methods such as energy spectra and helicity profiles. The electromotive force is one of the second-order fluctuation quantities (cf., the Reynolds stress tensors in fluid dynamics). Though the number of studies is limited, the properties of electromotive force are determined using the Helios spacecraft data in the inner heliosphere. To summarize, the properties are:

1. The electromotive force is non-zero even in the quiet solar wind. Its magnitude is of the order of mV km$^{-1}$ in the quiet solar wind, corresponding to the fluctuating flow velocity of 1 km s$^{-1}$ and the fluctuating magnetic field of 1 nT, and can reach the order of 1 or 10 V km$^{-1}$ during the magnetic clouds or shock crossings.

2. The fluctuations of electromotive force are nearly random, and the spectrum (in the spacecraft-frame frequency domain) represents a power-law curve with a slope close to $-5/3$.

3. The mean-field model of electromotive force can be tested against the Helios data. The proportionality does not hold between the electromotive force and the alpha effect, but together with the magnetic diffusion (beta term) and the cross helicity effect (gamma term) the electromotive force can qualitatively be reconstructed using the large-scale magnetic field and flow velocity.

4. The electromotive force during the shock crossings decays as a function of the radial distance from the Sun, from 1–10 V km$^{-1}$ at a distance of 0.3 AU down to 0.1–1 V km$^{-1}$.

Local magnetic field amplification is possible in the solar wind, and is associated with the electromotive force (in particular, the alpha and cross helicity effects). Crossing of coronal mass ejections or transient events or wake region behind obstacles such as planets or other solar system bodies (asteroids, satellites, and comets) may be potential regions of interest for testing for non-zero electromotive force.

Statistical behavior of turbulent fields is more complete when the electromotive force is properly assessed or modeled in addition to the energy densities (for the magnetic field and the flow velocity) and helicity quantities (for the cross helicity, current helicity, and kinetic helicity). It is important to note here that the construction of mean field and identification of fluctuating fields is not unique. The mean field is determined by smoothing (e.g., running average), local filter (boxcar, Gaussian), and low-pass filter. Since solar wind turbulence has fluctuations on various spatial and temporal scales, the magnitude of electromotive force may likely depend on the averaging process such as coarse graining.

The electromotive force can serve as a data analysis tool. Hofer and Bourdin (2019) proposed a classification scheme for the shock crossings into the jump type (e.g., coronal mass ejections) and the transient type (e.g., co-rotating interaction regions). Various types of discontinuities or structures may be better identified using the electromotive force, for example, shock types (fast, slow, and intermediate shocks), magnetic reconnection exhausts, detailed structures within the current sheets and shocks.

*Acknowledgements.* YN thanks many colleagues for stimulating discussions and providing data and materials on the concept of electromotive force: Philippe-A. Bourdin, Iver Cains, Abraham Chian, Tohru Hada, Bernhard Hofer, Gurbax Lakhina, Eckart Marsch, Donald Melrose, and Nobumitsu Yokoi. This work was financially supported by fthe Austrian Research Promotion Agency (FFG) under contract ASAP 865967.

*Code/Data availability.* Helios data are available at CDAWeb https://cdaweb.sci.gsfc.nasa.gov (NASA CDAWeb, 2021).

*Author contribution.* This is a single-author work.

*Competing interests.* The author declares no conflict of interest.

## Appendix:  Appendix: Second-order quantities

Energy densities, helicity densities, and electromotive force are the second-order fluctuation quantities using the magnetic field and flow velocity. The energy density of fluctuating magnetic field is

$$e_\mathrm{m} \;\; = \;\; \frac{1}{2\mu_0} \left( \langle b_\mathrm{x} b_\mathrm{x} \rangle + \langle b_\mathrm{y} b_\mathrm{y} \rangle + \langle b_\mathrm{z} b_\mathrm{z} \rangle \right). \tag{25}$$

The kinetic energy density is

$$e_\mathrm{k} \;\; = \;\; \frac{1}{2} \rho \left( \langle u_\mathrm{x} u_\mathrm{x} \rangle + \langle u_\mathrm{y} u_\mathrm{y} \rangle + \langle u_\mathrm{z} u_\mathrm{z} \rangle \right), \tag{26}$$

where $\rho$ is the mass density of medium.

The cross helicity density is a correlation between the magnetic field and the flow velocity,

$$h_\mathrm{crs} \;\; = \;\; \langle u_\mathrm{x} b_\mathrm{x} \rangle + \langle u_\mathrm{y} b_\mathrm{y} \rangle + \langle u_\mathrm{z} b_\mathrm{z} \rangle. \tag{27}$$

The current helicity density is

$$
\begin{aligned}
h_{\mathrm{crt}} &= \langle (\nabla \times \boldsymbol{b}) \cdot \boldsymbol{b} \rangle & (28) \\
&= \partial_{\mathrm{x}} \left( \langle b_{\mathrm{y}} b_{\mathrm{z}} \rangle - \langle b_{\mathrm{z}} b_{\mathrm{y}} \rangle \right) \\
&\quad + \partial_{\mathrm{y}} \left( \langle b_{\mathrm{z}} b_{\mathrm{x}} \rangle - \langle b_{\mathrm{x}} b_{\mathrm{z}} \rangle \right) \\
&\quad + \partial_{\mathrm{z}} \left( \langle b_{\mathrm{x}} b_{\mathrm{y}} \rangle - \langle b_{\mathrm{y}} b_{\mathrm{x}} \rangle \right). & (29)
\end{aligned}
$$

Note that the helicity in general (e.g., magnetic helicity density and current helicity) is non-zero when the field is helical, e.g., when choosing the left-handed (or right-hand) helical field around the mean field $B_0$ in the z direction,

$$
\begin{bmatrix} b_{\mathrm{x}} \\ b_{\mathrm{y}} \\ b_{\mathrm{z}} \end{bmatrix}
=
\begin{bmatrix} \delta b \exp\left( -kz \right) \\ \delta b \exp\left( -kz \pm \frac{\pi}{2} \right) \\ B_0 \end{bmatrix},
\tag{30}
$$

where the plus sign is for the left-hand helical field when tracking the field rotation sense along the wavevector in the z direction $k$, and the minus sign for the right-hand helical field, respectively. $\delta b$ denotes the amplitude of the helical rotation. The magnetic helicity density can also be constructed from the fluctuating magnetic field by un-curling the vector potential $\boldsymbol{A} = \nabla \times \boldsymbol{B}$ in the Fourier domain under the Coulomb gauge as

$$
\begin{aligned}
h_{\mathrm{mag}} &= \langle \boldsymbol{A} \cdot \boldsymbol{B} \rangle & (31) \\
&= \int \mathrm{d}^3 r \, \mathrm{e}^{\mathrm{i} \boldsymbol{k} \cdot \boldsymbol{r}} \\
&\quad \times \Bigg[ -\frac{\mathrm{i}}{k^2} \Big[ k_{\mathrm{x}} \left( \langle b_{\mathrm{y}}^* b_{\mathrm{z}} \rangle - \langle b_{\mathrm{z}}^* b_{\mathrm{y}} \rangle \right) \\
&\qquad + k_{\mathrm{y}} \left( \langle b_{\mathrm{z}}^* b_{\mathrm{x}} \rangle - \langle b_{\mathrm{x}}^* b_{\mathrm{z}} \rangle \right) \\
&\qquad + k_{\mathrm{z}} \left( \langle b_{\mathrm{x}}^* b_{\mathrm{y}} \rangle - \langle b_{\mathrm{y}}^* b_{\mathrm{x}} \rangle \right) \Big] \Bigg]. & (32)
\end{aligned}
$$

The kinetic helicity density is constructed as

$$
\begin{aligned}
h_{\mathrm{k}} &= \langle (\nabla \times \boldsymbol{u}) \cdot \boldsymbol{u} \rangle & (33) \\
&= \partial_{\mathrm{x}} \left( \langle u_{\mathrm{y}} u_{\mathrm{z}} \rangle - \langle u_{\mathrm{z}} u_{\mathrm{y}} \rangle \right) \\
&\quad + \partial_{\mathrm{y}} \left( \langle u_{\mathrm{z}} u_{\mathrm{x}} \rangle - \langle u_{\mathrm{x}} u_{\mathrm{z}} \rangle \right) \\
&\quad + \partial_{\mathrm{z}} \left( \langle u_{\mathrm{x}} u_{\mathrm{y}} \rangle - \langle u_{\mathrm{y}} u_{\mathrm{x}} \rangle \right). & (34)
\end{aligned}
$$

And the electromotive force is

$$
\boldsymbol{E}_{\mathrm{emf}}
=
\begin{bmatrix} \langle u_{\mathrm{y}} b_{\mathrm{z}} \rangle - \langle u_{\mathrm{z}} b_{\mathrm{y}} \rangle \\ \langle u_{\mathrm{z}} b_{\mathrm{x}} \rangle - \langle u_{\mathrm{x}} b_{\mathrm{z}} \rangle \\ \langle u_{\mathrm{x}} b_{\mathrm{y}} \rangle - \langle u_{\mathrm{y}} b_{\mathrm{x}} \rangle \end{bmatrix}.
\tag{35}
$$

From a data-analysis point of view, the second-order quantities introduced above can be derived from the correlation matrices (or spectral density matrices when working in the spectral domain):

$$\mathbf{M}_{bb} = \langle \boldsymbol{b}\,\boldsymbol{b}^{\mathrm{t}} \rangle \tag{36}$$

$$= \begin{bmatrix} \langle b_{\mathrm{x}}\, b_{\mathrm{x}} \rangle & \langle b_{\mathrm{x}}\, b_{\mathrm{y}} \rangle & \langle b_{\mathrm{x}}\, b_{\mathrm{z}} \rangle \\ \langle b_{\mathrm{y}}\, b_{\mathrm{x}} \rangle & \langle b_{\mathrm{y}}\, b_{\mathrm{y}} \rangle & \langle b_{\mathrm{y}}\, b_{\mathrm{z}} \rangle \\ \langle b_{\mathrm{z}}\, b_{\mathrm{x}} \rangle & \langle b_{\mathrm{z}}\, b_{\mathrm{y}} \rangle & \langle b_{\mathrm{z}}\, b_{\mathrm{z}} \rangle \end{bmatrix} \tag{37}$$

$$\mathbf{M}_{uu} = \langle \boldsymbol{u}\,\boldsymbol{u}^{\mathrm{t}} \rangle \tag{38}$$

$$= \begin{bmatrix} \langle u_{\mathrm{x}}\, u_{\mathrm{x}} \rangle & \langle u_{\mathrm{x}}\, u_{\mathrm{y}} \rangle & \langle u_{\mathrm{x}}\, u_{\mathrm{z}} \rangle \\ \langle u_{\mathrm{y}}\, u_{\mathrm{x}} \rangle & \langle u_{\mathrm{y}}\, u_{\mathrm{y}} \rangle & \langle u_{\mathrm{y}}\, u_{\mathrm{z}} \rangle \\ \langle u_{\mathrm{z}}\, u_{\mathrm{x}} \rangle & \langle u_{\mathrm{z}}\, u_{\mathrm{y}} \rangle & \langle u_{\mathrm{z}}\, u_{\mathrm{z}} \rangle \end{bmatrix} \tag{39}$$

$$\mathbf{M}_{ub} = \langle \boldsymbol{u}\,\boldsymbol{b}^{\mathrm{t}} \rangle \tag{40}$$

$$= \begin{bmatrix} \langle u_{\mathrm{x}}\, b_{\mathrm{x}} \rangle & \langle u_{\mathrm{x}}\, b_{\mathrm{y}} \rangle & \langle u_{\mathrm{x}}\, b_{\mathrm{z}} \rangle \\ \langle u_{\mathrm{y}}\, b_{\mathrm{x}} \rangle & \langle u_{\mathrm{y}}\, b_{\mathrm{y}} \rangle & \langle u_{\mathrm{y}}\, b_{\mathrm{z}} \rangle \\ \langle u_{\mathrm{z}}\, b_{\mathrm{x}} \rangle & \langle u_{\mathrm{z}}\, b_{\mathrm{y}} \rangle & \langle u_{\mathrm{z}}\, b_{\mathrm{z}} \rangle \end{bmatrix} . \tag{41}$$

The magnetic and kinetic energy densities correspond to the diagonal elements of $\mathbf{M}_{bb}$ and $\mathbf{M}_{uu}$, respectively. The cross helicity density is derived from the diagonal elements of $\mathbf{M}_{ub}$. The current and kinetic helicity densities are constructed from the off-diagonal elements of $\mathbf{M}_{bb}$ and $\mathbf{M}_{uu}$, respectively. The electromotive force is constructed from the off-diagonal elements

of $\mathbf{M}_{ub}$. The Reynolds stress tensors for magnetohydrodynamic turbulence are constructed as $\mathbf{R}_{\mathrm{k}} = \mathbf{M}_{bb} - \mathbf{M}_{uu}$ for the kinetic variant and $\mathbf{R}_{\mathrm{m}} = \mathbf{M}_{ub} - \mathbf{M}_{bu}$ and the magnetic variant (Yoshizawa, 1990).

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
