# Peer review of "Electromotive force in the solar wind"

_Annales Geophysicae, 2021_

## Author Comment (AC1)

**Reply to referee comments**
Electromotive force in the solar wind
Manuscript ID: angeo-2021-18

Y. Narita
* * *
**Referee 1**

1. *The review article by Yasuhito Narita is on "Emf in solar wind" an important topic of relevance to both the Astrophysical dynamo and MHD turbulence community. It is well written and well-presented. The author has published other papers on this topic, one of which is cited but its results are not presented in detail.*

   **Reply**:

   - The main message is of Narita and Vörös (AnGeo 2018) was missing in the original manuscript. The message is that the transport coefficients alpha and beta can be estimated using analytic form under some assumptions. Moreoever, the solar wind event studied by Narita and Vörös was revisited to deepen the study of electromotive force to improve the quality of the manuscript. The coefficients alpha and beta are observationally evaluated as a function of the fluctuating flow speed and also as a function of the fluctuating magnetic field. The result is shown in Figure 4 and in section 3.3.2. The second half of section 3.3.2 is an original and novel contribution of the author to the studies of electromotive force. For this reason, "Review: " was deleted from the title.

   - The following text was added (page 7, line 157 to page 8, line 184).

     "For the simple model with the alpha and beta terms (indicating the field amplification and the turbulent diffusion, respectively), analytic forms are proposed to estimate the transport coefficients alpha and beta (Narita and Vörös, 2018). For this purpose we model the electromotive force in the following form:

     $$\vec{E}_{\mathrm{emf}} = \alpha \langle \vec{B} \rangle - \beta \nabla \times \langle \vec{B} \rangle \qquad (15)$$

     . Vector product between the mean magnetic field $\langle \vec{B} \rangle$ and the electromotive force $\vec{E}_{\mathrm{emf}}$ in Eq. (15) yields

     $$\langle \vec{B} \rangle \times \vec{E}_{\mathrm{emf}} = -\beta \langle \vec{B} \rangle \times (\nabla \times \langle \vec{B} \rangle), \qquad (16)$$

     which can be arranged into an estimator for the beta coefficient as

     $$\beta = \frac{1}{F^2} \vec{F} \cdot (\langle \vec{B} \rangle \times \vec{E}_{\mathrm{emf}}). \qquad (17)$$

     Here, $\vec{F}$ denotes the Lorentz force for the large-scale magnetic field and is defined as (by neglecting the permeability of free space $\mu_0$)

     $$\vec{F} = (\nabla \times \langle \vec{B} \rangle) \times \langle \vec{B} \rangle. \qquad (18)$$

For the coefficient alpha we multiply Eq. (15) by the maen magnetic field $\langle \vec{B} \rangle$ and obtain:

$$\langle \vec{B} \rangle \times \vec{E}_{\text{emf}} = \alpha(\langle \vec{B} \rangle)^2 - \beta \langle \vec{B} \rangle \cdot (\nabla \times \langle \vec{B} \rangle). \tag{19}$$

Equation (19) can be arranged into:

$$\alpha = \frac{1}{\langle \vec{B} \rangle^2} \left[ \langle \vec{B} \rangle \cdot \vec{E}_{\text{emf}} + \frac{H_{\text{C}}}{F^2} \vec{F} \cdot (\langle \vec{B} \rangle \times \vec{E}_{\text{emf}}) \right], \tag{20}$$

by using the estimator for the coefficient beta (Eq. 17) and introducing the (density of) large-scale current helicity $H_{\text{C}}$ as

$$H_{\text{C}} = (\nabla \times \langle \vec{B} \rangle) \cdot \langle \vec{B} \rangle. \tag{21}$$

The coefficients alpha and beta are evaluated observationally using Eqs. (17) and (20), and graphically plotted as functions of the fluctuating flow speed $u = |\vec{u}|$ and fluctuating magnetic field $b = |\vec{b}|$ on the logarithmic scale in Fig. 4. The coefficients alpha and beta exhibit the following properties:

(a) The both coefficients are scattered to a larger extent over the flow speed flucutation $u$ and the magnetic field fluctuation $b$. Variation of the coefficient alpha spans from $10^{-4}$ km s$^{-1}$ to $10^4$ km s$^{-1}$ (8 orders of magnitude), and that of beta span from $10^6$ km$^2$ s$^{-1}$ to $10^{16}$ km$^2$ s$^{-1}$ (10 orders of magnitude).

(b) Yet, the both coefficients show a systematic trend that the values of coefficients increase at larger fluctuation amplitudes. The systematic trend appears not only in the flow speed domain (left panels) but also in the magnetic field domain (right panels). The systematic trend may as well be (observationally) modeled using a power-law scaling (linearly on the logarithmic scale).

[Figure]

Fig. 4    Transport coefficients alpha and beta as functions of the fluctuating flow speed and fluctuating magnetic field for the two-component electromotive force model with the alpha and beta terms. The Helios solar wind data and the transport coefficients studied by Narita and Vörös (2018) are used for the graphics."

2. *In my view, the paper will benefit from more depth and the breadth, being a review article. It would be relevant to get a glimpse of the entire body of work done in the area and allied areas (unless there is a page limit). For e.g. the classical works or review articles by Paul Roberts on dynamo theory, Keith Moffatt on helicity, for example, are not cited. There are hardly 10 referenes!*

   **Reply**:

   - Agreed. Review articles about the dynamo mechanism and the related subjects were added in the revision (section 1 and section 2). The number of references is 43 in the revision.

   - The follwing text was added (page 1, line 10–19).

     "...magnetic fields in turbulent fluid motions (Elsasser, 1956; Moffatt, 1978; Roberts and Soward, 1992). Examples of large-scale magnetic field generation associated with the dynamo mechanism can be found in geophysical, solar system, and astrophysical applications such as Earths magnetic field (Glatzmaier and Roberts, 1998; Glatzmaier, 2002; Roberts and Glatzmaier, 2000; Kono and Roberts, 2002), planetary magnetic fields (Jones, 2011), Jupiters moon (Ganymede) intrinsic field (Schubert et al., 1996; Sarson et al., 1997), solar magnetic field (Charbonneau, 2010, 2014; Brandenburg, 2018), stellar magnetic fields (Berdyugina, 2005; Brun and Browning, 2017), and galactic and extragalactic fields (Vainshtein and Ruzmaikin, 1971; Kronberg, 1994; Widrow, 2002; Beck et al., 2020). Our understanding of the dynamo mechanism is being deepened and broadended by using numerical simulations using the fundamental equations and analytic treatment and modeling (Brandenburg, 2018). Recent theoretical study by Yokoi (2018a) suggests that the electromotive force and the density variation are locally enhanced such as in the shock-front region, and the density enhancement would lead to a fast magnetic reconnection."

3. *A specific comment on one of the conclusions – why is it so that the emf can be reconstructed better by considering the beta and gamma terms. Which of these two is contributing more? The author being an expert in this area could shed some light on this interesting conclusion. If this is difficult to understand, then the author should say so along with the reasons.*

   **Reply**:

   - I agree with the importane of the question raised by the referee, and undertook a further analysis of the reconstruction work by Bourdin et al. (ApJ 2018). The three terms (alpha-term, beta-term, and gamma-term) are found to be of the same order, indicating that the cross helicity dynamo effect (represented by the gamma term) should not be neglected in the inner heliosphere. Moreover, this cross helicity term may be the leading term when the solar wind arrives at the Earth orbit (1 AU) and the magnetic field becomes weeker by a factor of about 10 from the field in the inner heliosphere. A paragraph was added in section 3.3.3.

- The following text was added (page 9, line 199 to page 10, line 213).

"It is interesting to compare among the three terms in the electromotive force model (alpha-term, beta-term, and gamma-term) in Eq. (5) using the order-of-magnitude estimate method. The reconstruction work by Bourdin et al. (2018) determined the values of coefficients alpha, beta, and gamma as shown in Tab. 1.

Tab. 1 Transport coefficients estimated from a 12-hour solar wind interval including an interplanetary shock (active solar wind) and a quasi-stationary turbulent state (quiet solar wind) after Bourdin et al. (2018).

| coefficient | active solar wind | quiet solar wind |
|:---:|:---:|:---:|
| $\alpha$ | $-50 \mathrm{~km\,s^{-1}}$ | $\pm 5 \mathrm{~km\,s^{-1}}$ |
| $\beta$ | $50 \times 10^6 \mathrm{~km^2\,s^{-1}}$ | $5 \times 10^6 \mathrm{~km^2\,s^{-1}}$ |
| $\gamma$ | $-10 \times 10^6 \mathrm{~km\,nT}$ | $\pm 1 \times 10^6 \mathrm{~km\,nT}$ |

The ratio of the alpha term (helical dynamo term) to the beta term (turbulent diffusion term) is estimated nearly of the order of unity:

$$\frac{|\alpha \langle \vec{B} \rangle|}{|\beta \nabla \times \langle \vec{B} \rangle|} \sim \frac{\alpha L}{\beta} \sim 4 \tag{22}$$

where the spatial gradient scale is estimated about $L = 4 \times 10^6$ km in the solar wind corresponding to a Doppler-shifted frequency of about $10^4$ Hz (e.g., Tu and Marsch, 1995). The order-of-unity estimate as in Eq. (22) is valid for both the active solar wind and the quiet solar wind when referring to the observational values of the transport coefficeints in Tab. 1. The ratio of the gamma term (cross helicity term) to the beta term is estimated of the order of unity, too:

$$\frac{|\gamma \nabla \times \langle \vec{U} \rangle|}{|\beta \nabla \times \langle \vec{B} \rangle|} \sim \frac{\gamma B_0}{\beta U_0} \sim 2. \tag{23}$$

Here we used a flow speed of $U_0 = 400 \mathrm{~km\,s^{-1}}$ (typical both in the inner heliosphere and around the Earth orbit) and a magnetic field of $B_0 = 40$ nT (typical in the inner heliosphere but not around the Earth). The cross helicity term plays a more important role because the flow speed does not change very much over the radial distances from the Sun while the magnetic field decays radially due to the flux conservation over the spatial expansion. Around the Earth orbit, the ratio of the gamma term to the beta term is expected about 10 times larger than that in the inner heliosphere."
* * *
**Referee 2**

1. *This paper is well written and covers an important topic. This (mini) review gives a clear indication that the study of the EMF in the solar wind is a valuable*

*activity (and I agree with the author on this). The mean induction equation, with the alpha, beta and gamma split, is presented in equation (5). Shortly after, different expressions are given for alpha and beta, and then gamma. Although there are similarities between the expressions, the assumptions going into deriving those terms are very different (compare Krause and Radler with Yokoi). Some more information on the applicability of these models for the solar wind would be helpful.*

**Reply**:

- The difference in the derivations is emphasized in the revision (page 4, line 95–100)

  "It is worth noting that the assumptions in the derivation of the transport coefficients are different between Eqs. (6)–(7) and Eqs. (9)–(11); the former expressions are based on homogeneous turbulence in a rotating flow, while the lattere expressions are based on the response function (Green function) of inhomogeneous turbulence. Extension of Eq. (6) to Eq. (9) indicates that the residual helicity between the kinetic helicity and the current helicity drives the dynamo effect (Pouquet et al. 1976). The importance of the cross helicity term (with the coefficient gamma) has largely been overlooked in the earlier studies because the large-scale flow velocity was eliminated by using the Galilean invariance."

- Appicability of the cross helicity in the solar wind is mentioned, too (page 4, line 108–112).

  "The cross helicity effect may play an important role in the solar wind, as the cross helicity can be interpreted as the energy difference between two counter-propagating Alfvén wave packets when using the Elsässer variables, and is expected to evolve in the solar wind over the heliocentric distances if the Alfvén waves are excited near the Sun, propagate unidirectionally (away from the Sun) in the inner heliosphere, and gradually undergoes scatterings or instabilities to excite backward-propagating Alfvén waves."

2. *The author then highlights solar wind studies which use various forms of the EMF expression and discusses how effective they have been. What he has written is very clear.*

   *I think there could be some more discussion on the physical importance of the alpha, beta and gamma effects for the solar wind.*

**Reply**:

- Discussed in section 2 (page 4, line 101–112).

  "Transport of the kinetic helicity and tha current helicity (or magnetic

helicity) from the solar convection zone to the heliosphere remains one of the open questions. The spacecraft observations indicate that magnetic helicity changes the sign nearly randomly over the spacecraft-frequencies. However, as discussed in section 3, the alpha effect may locally be enhanced when a transient event (e.g., coronal mass ejections) passes by.

Diffusion of large-scale magnetic field by the beta term is expected to be persistenly large in the solar wind, considering the fact that the solar wind exhibits sign of developed or fully-developed turbulence with power-law energy spectra for the flow velocity and the magnetic field.

The cross helicity effect may play an important role in the solar wind, as the cross helicity can be interpreted as the energy difference between two counter-propagating Alfvén wave packets when using the Elsässer variables, and is expected to evolve in the solar wind over the heliocentric distances if the Alfvén waves are excited near the Sun, propagate unidirectionally (away from the Sun) in the inner heliosphere, and gradually undergoes scatterings or instabilities to excite backward-propagating Alfvén waves."

3. *I don't think the appendix is necessary and could be replaced, without much loss. For example, the author includes magnetic helicity in the appendix, but this does not really feature in the main text.*

    **Reply**:

    - From the observational point of view, the electromotive force is treated equally to the other second-order quantities, and I find that the readers benefit from the appendix section. I added the following text in the revision to emphasize the second-order quantities (page 3, line 57–65).

        "The electromotive force can be observationally determined when the flow velocity data and the magnetic field data are are available. In general, it is convenient to determine the covariance matrices for the magnetic field as $\mathbf{M_{bb}}$, for the flow velocity as $\mathbf{M_{uu}}$, and for the cross correlation between the flow velocity and the magnetic field as $\mathbf{M_{ub}}$. The electromotive force is constructed from the off-diagonal elements of the cross correlation matrix $\mathbf{M_{ub}}$. The mean-field dynamo theory predicts that the electromotive force is related to the energy and helicity quantities. Magnetic energy corresponds to the diagonal elements of the matrix $\mathbf{M_{bb}}$, and the kinetic energy the diagonal elements of the matrix $\mathbf{M_{uu}}$. Magnetic helicity and current helicity are constructed from the off-diagonal elements of the magnetic field matrix $\mathbf{M_{bb}}$, and the kinetic helicity from the off-diagonal elements of the flow velocity matrix $\mathbf{M_{uu}}$. The cross helicity is constructed from the diagonal elements of the cross correlation matrix $\mathbf{M_{ub}}$. The appendix section shows the second-order quantities that are accessible to the spacecraft observations."

**Other changes**

1. Due to the new, original scientic work, the manuscript is treated not any more as a mini review but as a regular paper with an extended review.

2. The following reference items were added to the manuscript.

[revised manuscript text omitted]

---

## Author Response (AR2)

Reply, angeo-2021-18, revision 2

Electromotive force in the solar wind
Yasuhito Narita

Again, I thank the editor and the both referees for careful check and thoughtful comments.
Reply comments are given here.

==========================================================

Referee 1

> The author has changed the article from a mini-review to a regular article.
> More data, results and a table have been added which has enhanced the quality
> of the paper on an important topic. I have only three comments which
> in my view should be addressed before the paper is accepted, and few minor
> editorial corrections. These can be easily incorporated in the manuscript
> if the author thinks them to be relevant.
>
> Comments
> 1) Since this is now a regular article, the article needs some more modification
> to reflect this. For e.g. the abstract needs to change to reflect some of
> the (new) observations mentioned in "summary and outlook".
> As done in his previous article in Ann Geo, Narita & Voros (2018)
> the author could include some more details on the methods,
> for the completeness of this article and for the benefit of the readers.

Reply

The author consulted with the editors about the treatment of the manuscript (review or regular),
and came to the conclusion that the manuscript type be kept as a review article.
The reason of my earlier intension to change the article type into a regular one
was that some journals do not accept any original data or new results in the review.
I learned that AnGeo accepts that the review articles may contain some new results
as far as the the new results do not dominate the article. So, the author finds the contents
related to Narita and Voros (2018) are presented on an appropriate level (not dominating
the manuscript).

* page 1, line 2 (abstract), "A review of the electromotive force ..."
* page 1, line 25 (section 1), "a review of ..."

> 2) On page 9, the author says "note that the alpha effect test fails on one hand,
> and the mean-field model can qualitatively reproduce the observed electromotive force."
> This is an intriguing result. The scaling analysis that follows (eq. 22) says
> that the alpha term should be almost as important as the beta term
> (in fact, 4 times larger) which is as important as the gamma term.
> Isn't this contradictory to the fact that the alpha effect test (section 3.3.1, fig 3)
> fails, according to the author himself? The author could shed more light
> on this aspect if he agrees with the contradiction. Perhaps the method
> used for alpha-test needs a correction/modification or a caveat?

Reply

It is not a contradiction, but my writing was confusing and misleading.
The text was corrected by emphasizing what the two results suggest as a lesson.

* page 9, line 198-206 (section 3.3.3), The following text has been added.

"It is interesting to note that the test for the single alpha effect (i.e., proportionality
of electromotive force solely to the mean magnetic field without the beta and
the gamma effects) fails against the solar wind data after by Marsch and Tu (1992),
yet that the test for the model with the three terms including the alpha, beta,
and gamma terms successfully reproduces the measured electromotive force
after Bourdin et al. (2018). The scaling analysis using Eq. (22) indicates that
the alpha term should be almost as important as the beta term (in fact,
4 times larger) which is as important as the gamma term. Hence, the lesson is
that the simplest model using only the alpha term is not sufficient, and
that the magnetic diffusion and the cross helicity effect should be considered
as well in the electromotive force composition. Under which conditions
the alpha effect will dominate remains an observationally open question;
perhaps there is a dependence on, e.g., fast or slow solar wind, quieter or more
disturbed solar wind, association with transient events such as coronal mass
ejections and corotating interaction regions."

> 3) Since this a regular article, the author should more clearly demarcate
> what is new here with respect to Narita & Voros (2018). Adding some
> text in the introduction and summary/outlook sections would address this.
> One page 1, the author says "This article presents the emf studies in the
> solar wind in view of the current in situ observations in the inner heliosphere
> such as Parker Solar Probe (since 2018), Solar Orbiter (since 2020), and
> BepiColombo's cruising to the Mercury orbit (since 2018)". However,
> the data considered is not from any of these new missions, but from Helios,
> same as in Narita & Voros (2018).

Reply

As replied in the point 1 above, the manuscript should remain for a review article,
and the manuscript should serve as an appetizer for analyzing the data coming
from the novel solar wind measurements. The analysis of Parker Solar Probe,
Solar Orbiter, or BepiColombo cruise data will be presented elsewhere.

> Minor points
> 1) Equation (5) occurs twice.

Reply

Corrected.

* page 3, line 82. Equation number (6).

> 2) Typo/spelling errors in lines 16, 57, 101, 105, 168

Reply

Done.

* L16: page 1, line 16, "broadened"
* L57: page 3, line 58-59, "are available"
* L101: page 4, line 103, "the current helicity"
* L105: page 4, line 107, "persistently large"
* L168: page 8, line 170, "mean magnetic field"

> 3) Full stop missing at line 116 and few other places

Reply

Done.

* page 4, line 118, "fluctuation quantity."

> 4) Line 181 - "both the" instead of "the both"

Reply

Done.

* page 8, line 180, "Both the coefficients"
* page 8, line 183, "both the coefficients"

> 5) Lines 48-49, 62, 81 - "the second term the ...", "the third term the ..." ,
> etc needs to be changed to "the second term represents/is the ..." and so on.

Reply

* L48: page 2, line 49, "the second term represents"
* L49: page 2, line 49, "the third term represents"
* L63: page 3, line 63, "kinetic energy corresponds to"
* L81: page 3, line 83, "the coefficient \beta represents"

> Please check the draft for some additional typos that I may have missed.
> Nowadays there are many auto spell-check softwares available.

Reply

Done.

* page 8, line 180, "flow speed fluctuation"
* page 9, line 214, transport coefficients in Tab. 1

=============================================================

Referee 2

> The author has improved the paper based on the previous set of referee comments.
> After addressing the following minor corrections, I would recommend publication.
>
> line 48: "frozen-in of the large-scale magnetic field" does not really make
> sense in this context. This term describes the deformation of the large-scale
> magnetic field by the large-scale flow.

Reply

I keep the sentence as is, and add the referee's suggestion in a bracket as follows.

* page 2, line 48-49, "(strictly speaking, deformation of the large-scale
  magnetic field by the large-scale flow),"

> line 58: "it is convenient to determine", did you mean "define" here?

Reply

Changed as follows.

* page 3, line 59, "In general, in the observational studies, it is more practical
  to construct the covariance matrices..."

> line 165: why not just include mu_0, or normalize it to 1?

Reply

It is set to unity for simplicity.

* page 8, line 167-168, "by setting the permeability of free space \mu_0 to unity for simplicity

> line 169: there is a typo on the LHS of equation (19) - it should be B dot E not B cross E.

Reply

Right, thank you!

* page 8, line 171, cdot on LHS in Eq. (20).

> line 172: I would simplify "(density of) large-scale current helicity"
> as "large-scale current helicity density".

Reply

Agreed.

page 8, line 174, "large-scale current helicity density"

> line 227: "or the Reynolds stress tensors in magnetohydrodynamics".
> I'm not sure why this statement is necessary - the Reynolds stress is
> a fluid dynamical quantity and not directly related to MHD.

Reply

The concept of Reynolds stress is often introduced to the problems
in magnetohydrodynamics as a generalized quantity, but to be safe
with nomenclature, I changed the text as follows.

* page 1, line 235, "(cf., the Reynolds stress tensors in fluid dynamics)."

> line 267: for consistency, perhaps change H_C in the main text to h_crt.

Reply

Yes!

* page 8, line 173, "h_crt" on RHS in Eq. (21)
* page 8, line 175, "h_crt" on LHS in Eq. (22)

> line 271: please specify which helicity you refer to.

Reply

It is the helicity in general sense.

* page 13, line 283, "Note that the helicity in general (e.g., magnetic helicity density and current
helicity)"

> line 302: typo: t4he.

Reply

Oh, ... thank you.

* page 14, line 314, "the off-diagonal"

> Although the meaning of the text is clear, the written English could be
> improved throughout the manuscript.

I ran the spelling checker.

* page 8, line 180, "flow speed fluctuation"
* page 9, line 214, transport coefficients in Tab. 1